# A Non-Conventional Platinum Drug against a Non-Small Cell Lung Cancer Line

**DOI:** 10.3390/molecules28041698

**Published:** 2023-02-10

**Authors:** Jéssica D. Silva, Joana Marques, Inês P. Santos, Ana L. M. Batista de Carvalho, Clara B. Martins, Raquel C. Laginha, Luís A. E. Batista de Carvalho, Maria Paula M. Marques

**Affiliations:** 1Molecular Physical-Chemistry R&D Unit, Department of Chemistry, University of Coimbra, 3004-535 Coimbra, Portugal; 2Department of Life Sciences, University of Coimbra, 3000-456 Coimbra, Portugal

**Keywords:** anticancer, platinum complexes, polyamines, A549 lung cancer, microRaman, microFTIR

## Abstract

A dinuclear Pt(II) complex with putrescine as bridging polyamine ligand ([Pt_2_Put_2_(NH_3_)_4_]Cl_4_) was synthesized and assessed as to its potential anticancer activity against a human non-small cell lung cancer line (A549), as well as towards non-cancer cells (BEAS-2B). This effect was evaluated through in vitro cytotoxicity assays (MTT and SRB) coupled to microFTIR and microRaman spectroscopies, the former delivering information on growth-inhibiting and cytotoxic abilities while the latter provided very specific information on the metabolic impact of the metal agent (at the sub-cellular level). Regarding cancer cells, a major impact of [Pt_2_Put_2_(NH_3_)_4_]Cl_4_ was evidenced on cellular proteins and lipids, as compared to DNA, particularly via the Amide I and Amide II signals. The effect of the chelate on non-malignant cells was lower than on malignant ones, evidencing a promising low toxicity towards healthy cells.

## 1. Introduction

According to the World’s Health Organization, cancer is the second main cause of death worldwide (following cardiovascular disorders), lung cancer presenting the highest mortality rate in both sexes, mostly due to the fact that it is often diagnosed at an advanced stage of the disease. The standard treatment relies on chemotherapy, through combined administration of platinum agents with another third-generation drug such as paclitaxel, docetaxel, gemcitabine or vinorelbine [1,2,3,4].

Cisplatin (*cis*-Pt(NH_3_)_2_Cl_2_, Figure 1A) has been widely used as an anticancer drug since its introduction to the clinics in 1978, upon the serendipitous discovery of its cell growth-inhibiting activity by Rosenberg, in the late 1960s [5,6]. Despite its high efficiency in early phases of treatment of several types of cancers (e.g., solid tumors), it presents serious drawbacks namely acquired resistance, deleterious side effects (e.g., nephrotoxicity, hepatoxicity and neurotoxicity) and lack of selectivity against numerous malignancies including metastatic cancers [7,8,9,10].

In the last two decades, Pt(II) and Pd(II) complexes comprising more than one metal center (polynuclear) and linear polyamine bridging ligands have been developed, based on clinically used compounds such as cisplatin [11,12,13,14,15,16], aiming at an enhanced activity and decreased deleterious side effects. It has been suggested that, upon intracellular hydrolysis of their leaving ligands (e.g., chlorides), they interact with their main pharmacological target (DNA) via non-conventional bonds—since they allow the formation of intra- and interstrand long-range adducts (not available for the conventional mononuclear Pt-drugs (e.g., cisplatin))—leading to an enhanced and less-repairable therapeutic effect [9,15,17]. More recently, a new class of polycationic platinum chelates without hydrolysable moieties was developed by Farrell et al. [18,19,20,21,22]. Instead of covalently binding to DNA, these compounds impact on the nucleic acid through electrostatic interactions with the phosphate groups, the so-called phosphate clamps. These interactions occur between the positively charged amino groups of the complex and the phosphates’ oxygen atoms. This is a third drug mode of action (MOA) for metallodrugs, apart from the conventional ones: intercalation (anticancer antibiotics) or groove binding (drugs with hydrolysable ligands). These types of polycationic metal chelates are prone to cause a greater distortion of the DNA helix when compared to their counterparts with hydrolysable groups which interact covalently with the molecule [18,20,22]. Additionally, this specific MOA (through phosphate clamps) has been shown to lead to a different spectrum of clinical activity, targeting cellular entities beyond DNA such as proteins, enzymes, proteasomes or proteoglycans [16]. Furthermore, given their particular chemical features, the binding of non-hydrolysable Pt-complexes to other biomolecules during the pharmacokinetic phase (e.g., glutathione, serum albumin or metallothioneins) is not expected to be so extensive, which is expected to lead to a decreased toxicity (deleterious side-effects) and a lower acquired resistance [22,23,24]. Biogenic polyamines—putrescine, spermidine and spermine—are ubiquitous in all eukaryotic cells, being essential for cell proliferation and differentiation. These compounds have been used successfully by the authors as biocompatible polydentate ligands to form chelates with Pt(II) and Pd(II) cations [8,9,10,12,13,15,16,17]. Although these polyamines are known to be closely related to neoplastic growth and cancer progression [25,26,27,28,29], this effect is not observed upon coordination to a metal center. Instead, they allow the formation of stable complexes with intermediate-to-soft transition ions (through nitrogen binding) that may be transported into the cell through the specific polyamine channels [30,31], which is an advantage for attaining a high bioavailability of the potential drugs at their pharmacological targets (namely DNA).

A drug’s impact on the cellular metabolic profile may be assessed using a state-of-the-art vibrational spectroscopy approach, namely microFTIR (micro-Fourier transform infrared) and microRaman. These are complementary techniques with a high specificity and spatial resolution, which allow the analysis of heterogenous samples such as cells and tissues at a subcellular resolution, providing unmatched biochemical information, including maps that combine chemical and morphological data. It is possible to achieve a spectral “fingerprint” representing the main cellular constituents such as DNA, proteins and lipids [32,33,34,35].

The present study focuses on the antineoplastic activity of a dinuclear Pt(II) complex with the biogenic diamine putrescine (Put, H_2_N(CH_2_)_4_NH_2_) against human lung cell lines—both malignant (A549) and non-malignant (BEAS-2B). This compound ([Pt_2_Put_2_(NH_3_)_4_]Cl_4_, Figure 1B) does not contain hydrolysable ligands (the chlorides act only as counterions) and has an overall positive charge ([Pt_2_Put_2_(NH_3_)_4_]^4+^), thus being expected to affect the cellular constituents through a non-conventional interplay, probably via the formation of phosphate clamps. Apart from in vitro cytotoxicity studies, the impact of [Pt_2_Put_2_(NH_3_)_4_]^4+^ on the metabolic profile of the cells was assessed by microFTIR and microRaman spectroscopies.

## 2. Results and Discussion

### 2.1. Evaluation of Anticancer Activity

[Pt_2_Put_2_(NH_3_)_4_]Cl_4_ was firstly synthesized by Farrell et al. [11], as an intermediate product for another polynuclear complex ([{*trans*-PtCl_2_(NH_3_)}_2_H_2_N(CH_2_)_4_NH_2_]). The effect of [Pt_2_Put_2_(NH_3_)_4_]^4+^ on the cell growth and viability was presently assessed for the A549 human lung cancer cell line, by the MTT (3-(4,5-dimethylthiazol-2-yl)-2,5-diphenyltetrazolium bromide) and SRB (Sulforhodamine B) assays. Cisplatin was used for comparison purposes (as a positive control). Both [Pt_2_Put_2_(NH_3_)_4_]^4+^ and cisplatin showed a significant dose-dependent reduction on cell viability and proliferation at 48 and 72 h of drug exposure (Figure 2). However, the lowest IC_50_ values (half maximal inhibitory concentration) for both assays were observed for the dinuclear Pt-agent—an IC_50_ of 1.6 µM at 72 h in the MTT assay, and of 0.4 µM at 48 h in the SRB test (Table 1). This complex displayed a higher cytotoxic activity against this cell line than those reported for other inorganic metal complexes with Schiff base ligands—having Ni(II), Cu(II), Co(II), Ag(I), Pd(II), Pt(II), Cr(III) or Mn(II) as metal centers—which presented IC_50_ values ranging from 9.40 to 25.58 µM (IC_50_ = 18.88 µM for the Pt(II) complex) after 72 h of incubation time [36].

The toxicity of [Pt_2_Put_2_(NH_3_)_4_]^4+^ against the BEAS-2B human lung non-cancer cell line was also evaluated by the MTT and SRB assays (Figure 3), the IC_50_ value (30.9 µM) having been calculated from the MTT data, after 72 h of incubation time. For the other incubation points, in both assays, IC_50_ were higher than the maximum concentration tested (32 µM) (Table 1). The lower toxicity found for this complex towards non-tumor lung cells contrasts with the 7- to 13-fold higher toxicity profile reported for cisplatin against the same cell line (IC_50_ values ranging from 2.4 to 4.15 µM, at 72 h) [37,38,39], which is a highly promising result regarding a potential future clinical application of [Pt_2_Put_2_(NH_3_)_4_]^4+^—a high anticancer activity coupled to a low toxicity against healthy cells being desired.

### 2.2. Evaluation of Metabolic Impact

Following the promising results obtained from the biological assays for [Pt_2_Put_2_(NH_3_)_4_]^4+^, its metabolic impact on human lung cancer cells (A549 cell line) was evaluated by analyzing the spectral biochemical signatures of the treated and non-treated (control) cells, obtained by microFTIR and microRaman spectroscopies. In line with the results from the MTT and SRB assays, the cells, formalin-fixed, were probed after 72 h of exposure to a concentration of 2 µM of [Pt_2_Put_2_(NH_3_)_4_]^4+^ (for which the compound showed the highest cytotoxic effect).

Formalin is a routinely used chemical fixative, which maintains the cellular components similar to in in vivo conditions, forming crosslinks between the formaldehyde and the primary and secondary amine groups of cellular proteins. Although the disruption of lipid assemblies and conformational protein changes may lead to a reduction in the intensity of the corresponding FTIR and Raman vibrational bands, the overall molecular content is not significantly altered by this fixation method and it is still possible to obtain a spectral profile resembling to a live cell [15,40,41]. Furthermore, given the fact that this is a comparison study (drug-free versus drug-incubated cells) this effect may be disregarded. No formalin contamination was observed in the presently acquired spectra, its characteristic vibrational modes at 907, 1040 and 1490 cm^−1^ not being detected [15].

For each condition tested, a large number of data points were collected as single point spectra across the cell population, by both microRaman (ca. 300 spectra) and microFTIR (ca. 400 spectra), in order to obtain specific and reliable biochemical information (spectral biomarkers) allowing to differentiate treated from untreated cells. Hence, average spectra could be built, representative of either the control or the drug-exposed samples, minimizing the effect of the cell cycle profile (Figure 4).

Both microRaman and microFTIR spectra of A549 cells, prior to drug-administration, revealed a vibrational profile of the main cellular constituents coinciding with a healthy state (Figure 4), according to the literature [15,42,43,44]. Cellular integrity was verified mainly by the presence of bands from nucleic acids (ν(OPO)_backbone_ and ν_s_(PO_2_^−^) characteristic from the native B-DNA conformation, at 786 and 1096 cm^−1^, respectively), as well as from proteins (ν(C=O) Amide I/α-helix, at 1659 cm^−1^). Comparing the average spectra of the untreated and [Pt_2_Put_2_(NH_3_)_4_]^4+^-treated samples, it was clear that the A549 cells were sensitive to the presence of the Pt(II) complex. However, since FTIR and Raman spectra of these highly heterogeneous biological samples can be complex and difficult to interpret due to their similarity (in this case from the same cell line), it is important to apply multivariate analysis techniques, such as principal component analysis (PCA), to identify patterns in the data and to interpret the spectral information that differs between the data groups. A simple analysis of the mean spectra might not completely reflect all the differences and variability found between the groups. Therefore, in order to identify specific features (biomarkers) representative of the biochemical effect of the Pt(II) agent and the physiological response from the cell, principal components analysis (PCA) of the spectroscopic data was performed.

### 2.3. Raman Microspectroscopy

Regarding microRaman, significant alterations were found in the fingerprint region of the spectrum (600 to 1800 cm^−1^) mainly along PC3, which represents 13.2% of the data explained variance (Figure 5A,D). These changes refer to nucleic acids, at specific vibrational modes: ν(OPO)_backbone_ at 789 cm^−1^ (higher in untreated samples) and 829 cm^−1^ (higher in drug-treated cells), respectively; ν(CC)_ring_ from adenine, cytosine, guanine and thymine (at 676 and 687 cm^−1^, higher in drug-treated cells), adenine and guanine from RNA (at 1301 cm^−1^, higher in the control), guanine (at 1557 cm^−1^) and adenine (at 1603 cm^−1^), both higher in drug-treated samples. Finally, features characteristic from the native DNA conformation (B-DNA) were visible in the PC3 loading plot (Figure 5D), namely ν(CC)_ring_ and δ(NH) vibrational modes from adenine, thymine and guanine, at 676 and 1647 cm^−1^ (both higher in drug-treated cells).

In the light of these results, other pharmacological targets may exist for this complex apart from DNA, since there were no significant conformational changes from B-DNA either to A-DNA or Z-DNA, upon drug exposure. Thus, this Pt(II) compound is expected to exert a different mode of action towards cancer cells that the one generally recognized for cisplatin: activation by hydrolysis of the chloride leaving groups, allowing its covalent binding to DNA (mainly at the nitrogen of the purine bases) [45,46,47]. Therefore, the positively charged complex [Pt_2_Put_2_(NH_3_)_4_]^4+^ has a different pharmacodynamic profile, since it does not have hydrolysable chloride ligands directly bound to the metal ions (the chloride counterions are in the outer coordination sphere of the complex): it is therefore prone to interact with the DNA double helix via electrostatic interactions between the positively charged amine groups of the ligand (putrescine), at physiological pH, and DNA’s negative phosphates, through a phosphate clamp type mechanism, such as that reported for Triplatin [22]. Indeed, the variation observed for DNA’s ν(OPO)_backbone_ vibrational modes in the drug-treated cells clearly revealed such an interplay, that has been suggested to trigger a distortion of the double helix [22,48]. Furthermore, the ν(CC)_ring_ and ν(CO) vibrations from RNA’s ribose and nucleotides suffered a drug-elicited change, along PC3, at 977, 1247, 1264 and 1124 cm^−1^, respectively.

When exposed to the dinuclear Pt-agent, changes at the protein level are anticipated, considering the cells’ drug resistance mechanisms, involving the binding of high mobility groups (proteins) to DNA, in order to try and repair the drug-triggered damage. A clear discrimination was observed along PC3 for ν_s_(CC) from phenylalanine’s ring at 1003 cm^−1^_,_ with a higher contribution in the treated cells. Further changes were observed in the proteins’ vibrational signature (along PC3), particularly: ν(CS) at 643 cm^−1^; ν_s_(CC) at 1557 cm^−1^ characteristic of tryptophan (both higher in drug-treated cells); ν(OPO) from phosphorylated proteins; ν(CC) at 1214 cm^−1^, from hydroxyproline, phenylalanine and tyrosine, higher in untreated cells; δ(CH_2_) and δ(CH_3_) at 1466 cm^−1^ (overlapping with CH_2_ and CH_3_ groups from aromatic lipids and CH_2_ from carbohydrates), higher in untreated cells; ν(C=C) from porphyrin at 1557 cm^−1^; ν(C=C) and ν(C=N) at 1586 cm^−1^; ν(C=C) and δ(NH_2_) from phenylalanine, tyrosine and tryptophan and δ(C=CH) from phenylalanine, at 1603 cm^−1^, all higher in drug-treated cells. Finally, ν_s_(CH_3_), Amide III (δ(CNH)+ν(NH)) and Amide I (ν(C=O)) from the α-helix conformation were discriminated according to PC3, respectively at 956, 1219 and 1647 cm^−1^ (higher in drug-treated cells), suggesting a drug binding to the membrane and to cytoplasmatic proteins, prompting variations in their primary and secondary structures. Furthermore, the impact of the drug on the cellular proteins was shown through differences (along PC3) in biomarkers associated to protein skeletal stretching modes, with a more significant contribution in drug-treated cells: ν_s_(CC)_ring_ from phenylalanine and tryptophan at 1003 and 1557 cm^−1^, respectively, and ν(C=O)_porphyrin_ at 1557 cm^−1^.

Concerning the lipids, there was a clear drug effect revealed by significant alterations in the drug-treated group: a higher contribution from ν(OPO)_phosphate esters_ at 687 cm^−1^, as well as from δ(CH) and ν_s_(PO_2_^−^) from phospholipids (at 1031 and 1088 cm^−1^, respectively, higher in drug treated cells); δ(CH_2_) (at 1150 and 1466 cm^−1^) and δ(CH_3_) from aromatic lipids at 1466 cm^−1^ (higher in untreated cells); ν(C=N) at 1586 cm^−1^ and ν(C=C) at 1586 and 1647 cm^−1^, both higher in the drug-treated cells. Cellular carbohydrates, in turn, displayed the worst discrimination between treated and untreated samples, the only vibrational modes with noteworthy changes being δ(CH_3_) from polysaccharides at 1466 cm^−1^, as observed in PC3.

The high wavenumber region (2700–3200 cm^−1^) provided a poor discrimination between the samples (Figure 5B,E). Nonetheless, some separation was still visible along PC2, representing 2.1% of the data explained variance. Analyzing the PC2 loading (Figure 5E) it was possible to notice changes in the bands assigned to the CH_2_ groups from proteins (ν_s_(CH_2_) at 2846 cm^−1^ and ν_as_(CH_2_) at 2930 cm^−1^, higher in the drug-treated cells), that unfortunately overlap with CH and CH_2_ groups from lipids and carbohydrates (ν_s_(CH) and ν_s_(CH_2_) at 2854 cm^−1^ and ν_as_(CH_2_) at 2934 cm^−1^) which may render these assignments less reliable.

### 2.4. FTIR Microspectroscopy

Regarding the microFTIR results, upon PCA analysis only the fingerprint region (900–1800 cm^−1^) allowed to distinguish both tested conditions (control and treated cells). The differences were mainly identified along PC5, representing 2% of the data variance (Figure 5C,F). The PC5 loading plot (Figure 5F) revealed alterations in specific features from nucleic acids, such as ν(CC)_ring_ from cytosine, guanine and thymine at 1170 cm^−1^ (higher in the drug-treated cells) and from guanine at 1344 cm^−1^, and δ(NH) vibrational mode of DNA, at 1655 cm^−1^. No major DNA conformational rearrangements were evidenced (e.g., in the helix backbone), although alterations in δ(NH) and in ν(CC)_ring_ from the nucleotides suggest that there is an impact on the nucleic acid, which is to be expected in the light of the potential drug interactions with the double-helix phosphate groups.

Similarly to the microRaman results, microFTIR data revealed that the most evident metabolic drug-induced changes occurred for the cellular proteins (Figure 4 and Figure 5C,F). Alterations were found, along PC5, for the vibrational modes: ν(O-CH_3_) and δ(CH)_phenylalanine_, at 1033 cm^−1^; ν(CC) and ν(CN), respectively at 1072 and 1153 cm^−1^; δ(CH) from tyrosine and phenylalanine, at 1170 cm^−1^. Moreover, when exposed to the drug CH_2_ deformations were also discriminated along PC5, at 1317 and 1394 cm^−1^. The most remarkable drug-induced changes were identified for Amide II, at 1552 cm^−1^, and for Amide I/α-helix, at 1655 cm^−1^.

The lipids also showed variations upon exposure to the Pt(II) complex, mainly in the phospholipids (from the membrane): ν_s_(OPO)_phospholipids_ at 1074 cm^−1^, ν_s_(NCH_3_)_phosphocoline_ at 1153 cm^−1^, δ(CH_2_) at 1317 cm^−1^, δ(CH_2_) at 1394 cm^−1^, all of these showing a higher contribution in the drug-treated cells, as observed along PC5. Once again, carbohydrates were found to be less affected by the drug, only the δ(CH_2_) vibrational modes at 1317 cm^−1^ having been observed to vary, overlapping with the corresponding signals from proteins and lipids.

The PCA analysis of the infrared data from the high wavenumber region (2800 to 3800 cm^−1^) did not allow to discriminate between both tested conditions (with and without drug).

## 3. Materials and Methods

### 3.1. Chemicals

Acetic acid glacial (99.7%), cisplatin (*cis*-Pt(NH_3_)_2_Cl_2_, >98%), diethyl ether, dimethyl sulfoxide (DMSO, ≥99.0%), 3-(4,5-dimethylthiazol-2-yl)-2,5-diphenyltetrazolium bromide (MTT), ethanol (99.8%), ethylenediaminetetraacetic acid (EDTA, disodium salt, dihydrate), formalin (10% neutral-buffered formalin, ca. 4% formaldehyde), Ham′s Nutrient Mixture F12 culture medium, methanol (≥99.8%), penicillin/streptomycin 100× solution, potassium chloride (≥99.5%), potassium phosphate monobasic (≥99.0%), putrescine (Put, 1,4-diaminobutane, ≥98.0%), Roswell Park Memorial Institute (RPMI) 1640 culture medium, sodium bicarbonate (NaHCO_3_, ≥99.0%), sodium chloride (99.0%), sodium phosphate dibasic (≥99.0%), Sulforhodamine B (SRB), trypan blue (0.4% *w*/*v*) and trypsin were purchased from Sigma-Aldrich Chemical S.A. (Sintra, Portugal). Fetal bovine serum (FBS) was obtained from Gibco-Life Technologies (Porto, Portugal).

### 3.2. Synthesis of the [Pt_2_Put_2_(NH_3_)_4_]Cl_4_ Complex

The [Pt_2_Put_2_(NH_3_)_4_]Cl_4_ complex was synthesized according to the procedure reported by Farrell et al. [11]. It was characterized by FTIR (Appendix A), Raman (Appendix A) and NMR (Appendix A) spectroscopies (results in agreement with the published data [11]).

FTIR selected bands (cm^−1^): 251 and 286 (δ_N-Pt-N_), 511 (ν_asN-Pt-N_), 801 and 828 (ν_C-C_), 979 (ν_C-C_), 1037 (ν_CN_), 1211 (t_CH2_), 1314 (t_CH2_, δ_sCNH_), 1346 (ω_CH2_, δ_CNH_), 1480 (δ_CH2_), 1589 (δ_NH2_), 2868 (ν_sCH2_), 2934 (ν_asCH2_), 3095 and 3157 (ν_sNH2_), 3406 (ν_asNH2_).

Raman selected bands (cm^−1^): 283 (δ_N-Pt-N_), 603 (δ_N-C-C_), 1052 (ν_NC_, ν_CC_), 1071 (δ_PtN–H_, t_CH2_), 1311 (t_CH2_, ω_CH2_) and 1468 (δ_CH2_).

^1^H NMR (400 MHz, D_2_O): δ (ppm) = 2.73 (8H, s, HN_2_-C*H*_2_-CH_2_-CH_2_-C*H*_2_-NH_2_), 1.71 (8H, s, HN_2_-CH_2_-C*H*_2_-C*H*_2_-CH_2_-NH_2_).

### 3.3. Cell Culture and Biological Assays

The BEAS-2B human bronchial epithelial and the A549 human lung cancer cells lines were purchased from the American Type Culture Collection (ATCC, VA, USA). The A549 cells were grown as a monolayer in 75 cm^2^ cell culture flasks, in RPMI 1640 cell culture medium supplemented with 2.0 g/L of NaHCO_3_, 10% (*v*/*v*) inactivated FBS and 1% (*v*/*v*) penicillin-streptomycin, at 37 °C in a humidified atmosphere with 5% of CO_2_. They were sub-cultured twice a week (doubling time = 22 h [49]). The BEAS-2B cells were grown as a monolayer in 75 cm^2^ cell culture flasks, in Ham′s Nutrient Mixture F12 cell culture medium supplemented with 2.5 g/L of NaHCO_3_, 10% (*v*/*v*) inactivated FBS and 1% (*v*/*v*) penicillin-streptomycin, at 37 °C in a humidified atmosphere with 5% of CO_2_. They were sub-cultured twice a week (doubling time = 26 h [50]).

The cells were harvested upon addition of trypsin/EDTA solution, seeded in 96-well plates (15 × 10^3^ cells/cm^2^) and incubated at 37 °C. After allowing the cells to adhere for 24 h, different concentrations of [Pt_2_Put_2_(NH_3_)_4_]Cl_4_ (1.9 mM stock solution in 10% PBS in water) were added. No drug was added to the control group (substituted by an equal amount of solvent). The cells were further incubated for 24, 48 and 72 h, previously to the growth-inhibition and viability assays. Cisplatin (1.0 mM stock solution in PBS) was used as a positive control.

After each incubation period (24, 48, and 72 h), the cell viability and cell proliferation were evaluated by the MTT [51] and Sulforhodamine B (SRB) [52] methods, respectively. For the former, after medium removal, an MTT solution (0.5 mg/mL in cell culture medium) was added to each well. After 2 h of incubation at 37 °C, the MTT containing medium was removed and 100 µL of DMSO were added in order to dissolve the purple formazan crystals formed by MTT reduction within the viable cells. The optical density of the solutions was then measured at 550 nm. Concerning the SRB assay, the cells were washed with PBS and Mili-Q water and fixed with 1% (*v*/*v*) acetic acid in methanol, overnight, at −20 °C, followed by the addition of 100 µL SRB solution. The cells were then incubated at 37 °C for 1 h, thoroughly washed with 1% acetic acid (*v*/*v*) and dried at room temperature. The dye was solubilized with TRIS (10 mM, pH = 10) and the optical density measured at 560 nm.

### 3.4. Sample Preparation for Spectroscopic Analysis

After harvesting upon addition of trypsin/EDTA solution, the cells were centrifuged and the pellet was resuspended in culture medium and seeded at a concentration of 15 × 10^3^ cells/cm^2^, on optical substrates suitable for either FTIR or Raman acquisition (CaF_2_ or MgF_2_, respectively). Upon 24 h of incubation at 37 °C, in a humidified atmosphere with 5% of CO_2_, the cells were treated with [Pt_2_Put_2_(NH_3_)_4_]^4+^-2 µM according to the respective 50% cell growth inhibition value (IC_50_). After 72 h of drug exposure, the culture medium was removed, the cells were washed with 0.9% NaCl (*w*/*v*) and fixed with 4% formalin (diluted in 0.9% NaCl from the commercial neutral buffered formaldehyde solution). After 10 min, the disks were thoroughly washed with Mili-Q water and air-dried. Two replicates were prepared for each condition—treated and untreated (control) cells.

### 3.5. Spectroscopic Measurements

The ^1^H-NMR spectrum was recorded at room temperature, in a Bruker Advance III NMR spectrometer 400 MHz (Bruker, Billerica, USA). Chemical shifts are quoted in parts per million (ppm) from the residual protic solvent signal (D_2_O: ^1^H 4.79 ppm).

FTIR-ATR spectra (both far- and mid-IR) were collected using a Bruker Vertex 70 FTIR spectrometer (Bruker Optik GmbH, Ettlingen, Germany) purged by CO_2_-free dry air and a Bruker Platinum ATR single reflection diamond accessory. A liquid nitrogen-cooled wide band mercury cadmium telluride (MCT) detector and a Ge on KBr substrate beamsplitter were used for the mid-IR interval (400–4000 cm^−1^). A room temperature deuterated L-alanine-doped triglycine sulfate (DLaTGS) detector with a polyethylene window and a Si beamsplitter were used for the far-IR range (50–600 cm^−1^). One hundred and twenty-eight scans were summed for each spectrum, at 2 cm^−1^ resolution, applying the 3-term Blackman–Harris apodization function, yielding a wavenumber accuracy above 1 cm^−1^.

The microFTIR spectra were acquired for the mid-IR interval using the Bruker Optics Vertex 70 spectrometer (Bruker Optik GmbH, Ettlingen, Germany) coupled to a Bruker Hyperion 2000 microscope (Bruker Optik GmbH, Ettlingen, Germany), in transmission mode, with a 15× Cassegrain both condenser and objective and liquid nitrogen cooled MCT detector (Bruker Optik GmbH, Ettlingen, Germany). Each spectrum corresponded to the average of 128 scans (for both sample and background), at a 4 cm^−1^ resolution.

The microRaman spectra were collected in a WITec Raman microscope (WITec GmbH, Ulm, Germany) system alpha300R, coupled to an ultra-high throughput spectrometer 300 VIS grating (f/4 300 mm focal distance, 600 groves per millimeter blazed for 500 nm). The detection system was 1650 × 200 pixels thermoelectrically cooled (−55 °C) charge-coupled device camera, front-illuminated with NIR/VIS antireflection coating, with a spectral resolution < 0.8 cm^−1^/pixel. The spectrum of the pure complex [Pt_2_Put_2_(NH_3_)_4_]Cl_4_ was acquired using a diode laser of 785 nm as the excitation source, with 30 s integration time and 30 accumulations. The spectra of the cells were acquired using as excitation radiation a 532 nm line of a diode-pumped solid-state laser (ca. 29 mW at the sample position). Each spectrum was acquired with 10 s integration time and 5 accumulations, with a 100× Zeiss Epiplan objective (NA 0.80, WD 1.3 mm) (Carl Zeiss GmbH, Jena, Germany).

The optical density (absorbance) values of the samples from the MTT and SRB assays were measured in a BioTek μQuant MQX200 Microplate (Agilent, CA, USA) spectrophotometer, equipped with the Gen5 1.11.5 software (Agilent, CA, USA).

### 3.6. Statistical Analysis

The results from the biological assays were represented as a mean and standard error of the mean (SEM) and were obtained from at least four independent experiments (n = 4) each of which included four replicate measurements, carried out for each concentration of each tested complex plus the untreated control.

For the determination of the IC_50_ values of both [Pt_2_Put_2_(NH_3_)_4_]^4+^ and cisplatin, the results were fitted using nonlinear regression analysis, in sigmoidal dose–response curves (variable slope).

### 3.7. Data Preprocessing and Analysis

FTIR spectra were acquired for each of the two replicates of [Pt_2_Put_2_(NH_3_)_4_]^4+^-treated cells, as well as for the control. The OPUS 8.1 software was used for pre-processing the spectra: interferences from the atmospheric CO_2_ were compensated, and the intensity variations of the background were corrected within the interval 400–4000 cm^−1^. The spectra were further corrected to resonant Mie scattering by extended multiplicative signal correction function with 20 iterations [53,54]. The data were cropped to the 1000–1800 cm^−1^ range and vector-normalized. Based on the intensity of the characteristic band of Amide I, the quality of the spectra was ensured, the spectra with poor signal-to-ratio (Amide I band <1.0) having been discarded.

Raman spectra were acquired for the two replicates of Pt_2_-treated cells and the control, using the WITec Control FIVE 5.1 software (WITec GmbH, Ulm, Germany). The cosmic-ray spikes observed in the spectra were removed. Initial pre-processing was based on a PC-based noise reduction algorithm, retaining 20 principal components and then recombining the dataset. Spectra were further vector-normalized and filtered using a Savitzky-Golay function (window width of 5 points).

### 3.8. Multivariate Data Analysis

For both the microRaman and the microFTIR data, a principal components analysis (PCA) was performed, using the Quasar 1.5 software [55,56], with a view to analyze the biochemical effects of [Pt_2_Put_2_(NH_3_)_4_]^4+^ within the cells. Both scores and loading plots were obtained, allowing the identification of specific spectral biomarkers, relative to the control sample, that enabled to distinguish between drug-treated and untreated cells.

## 4. Conclusions

In this work, [Pt_2_Put_2_(NH_3_)_4_]^4+^ was evaluated as a novel potential anticancer agent against A549 lung cancer. Despite its synthetic procedure having already been reported, to the authors’ best knowledge this is the first study regarding its biological properties, particularly its anticancer capacity. The in vitro evaluation of the antiproliferative and cytotoxic activities of [Pt_2_Put_2_(NH_3_)_4_]^4+^ was performed against the A549 human non-small cell lung cancer as well as towards the non-tumorigenic lung cells BEAS-2B, using the standard biological assays SRB and MTT, respectively. The complex showed a very promising cytotoxic activity at 48 and 72 h incubation periods against the A549 cells, similar to cisplatin (taken as a positive control) or even better (under specific conditions). Furthermore, the impact on healthy cells was found to be lower when compared to the cancer ones, and also significantly decreased relative to previously reported data for cisplatin against BEAS-2B cells.

The metabolic impact of the Pt(II) complex on human lung cells was also determined, using state-of-the-art microFTIR and microRaman techniques which allowed to identify alternative targets to the expected one (DNA) for the conventional platinum agents (such as cisplatin). Interestingly, a major impact of [Pt_2_Put_2_(NH_3_)_4_]^4+^ was evidenced on cellular proteins and lipids, as compared to DNA.

The results presently gathered are quite promising regarding the search for new metallodrugs and pharmacological targets against A549 non-small cell lung cancer, which may help to circumvent drug deleterious side effects and resistance mechanisms that currently hinder the long-term administration of numerous anticancer drugs.

## Figures and Tables

**Figure 1 molecules-28-01698-f001:**
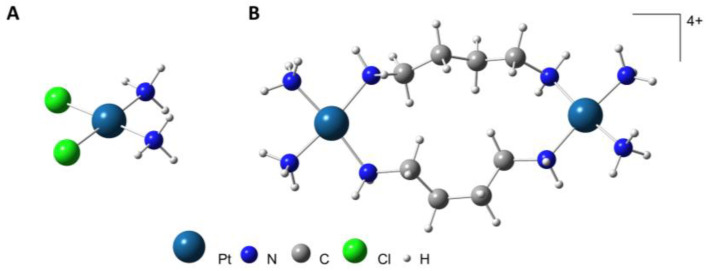
Schematic representation of the complexes cisplatin (**A**) and [Pt_2_Put_2_(NH_3_)_4_]^4+^ (**B**).

**Figure 2 molecules-28-01698-f002:**
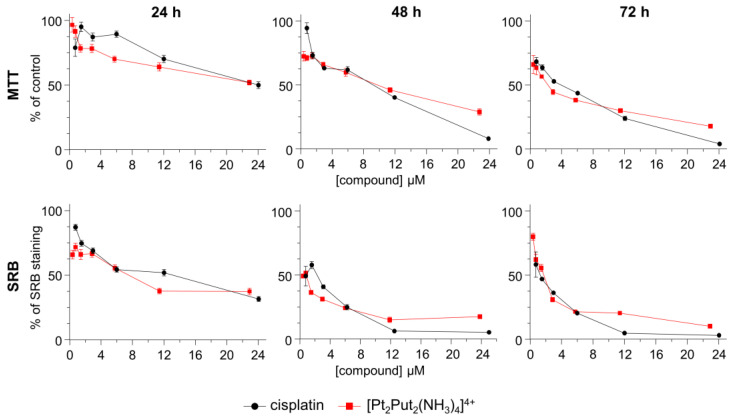
Dose–response curves for cisplatin and [Pt_2_Put_2_(NH_3_)_4_]^4+^ against the A549 human lung cancer cell line, at 24, 48 and 72 h incubation times. Data are expressed as mean ± standard error of the mean (SEM), *n* = 4.

**Figure 3 molecules-28-01698-f003:**
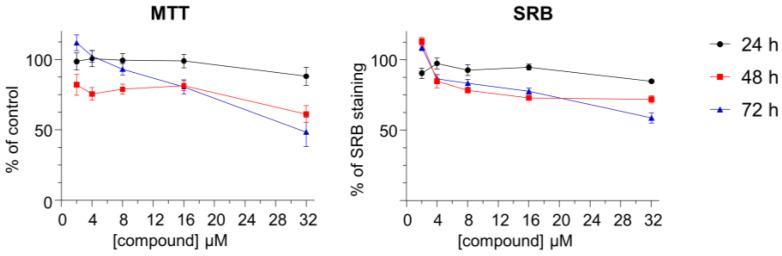
Dose–response curves for [Pt_2_Put_2_(NH_3_)_4_]^4+^ against the BEAS-2B human lung non-cancer cell line, at 24, 48 and 72 h incubation times. Data are expressed as mean ± SEM, *n* = 4.

**Figure 4 molecules-28-01698-f004:**
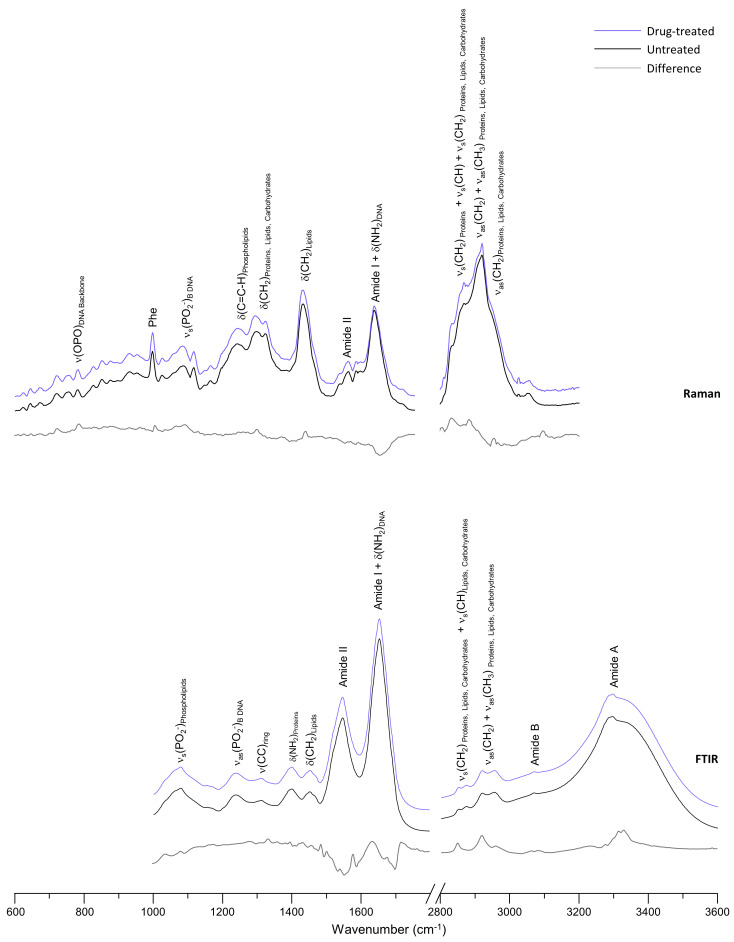
Mean Raman (600–3200 cm^−1^) and FTIR (1000–3600 cm^−1^) spectra for untreated (control) and [Pt_2_Put_2_(NH_3_)_4_]^4+^-treated A549 cells. The respective difference spectra to the control (drug-treated minus untreated) are also plotted.

**Figure 5 molecules-28-01698-f005:**
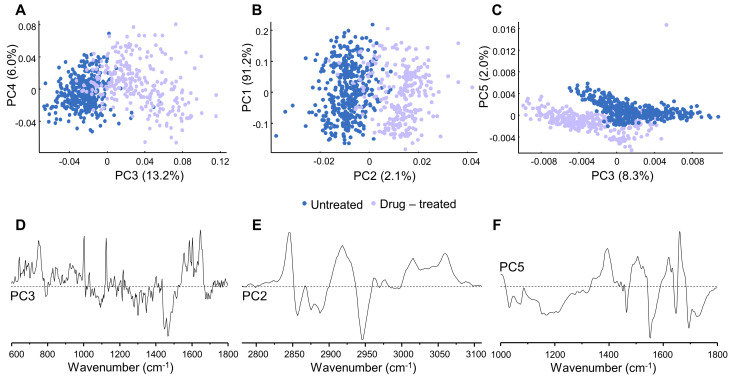
PCA scores and loading plots of Raman ((**A**,**D**) 600–1800 cm^−1^ and (**B**,**E**) 2700–3200 cm^−1^) and FTIR ((**C**,**F**) 900–1800 cm^−1^) data for [Pt_2_Put_2_(NH_3_)_4_]^4+^-treated lung cancer cell line (A549) vs. control (untreated cells). For clarity the loadings are offset, the dashed horizontal lines indicating zero loading.

**Table 1 molecules-28-01698-t001:** Half maximal inhibitory concentration (IC_50_, µM) of cisplatin and [Pt_2_Put_2_(NH_3_)_4_]^4+^ against the A549 human lung cancer and the BEAS-2B human lung non-cancer cell lines, at 24, 48 and 72 h incubation times.

Cell Line	Incubation Time (Hours)	Cisplatin		[Pt_2_Put_2_(NH_3_)_4_]^4+^	
		MTT	SRB	MTT	SRB
A549	24	32.9 ± 1.7	8.2 ± 1.1	>24	5.4 ± 1.3
	48	5.8 ± 1.2	1.2 ± 1.3	6.3 ± 1.3	0.4 ± 1.2
	72	2.6 ± 1.2	1.2 ± 1.1	1.6 ± 1.1	1.4 ± 1.1
BEAS-2B	24	n.d.	n.d.	>32	>32
	48	n.d.	n.d.	>32	>32
	72	n.d.	n.d.	30.9 ± 1.1	>32

Data from four independent measurements (n = 4); n.d.—not determined.

## Data Availability

The data presented in this study are available on request from the corresponding author.

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
