# Peer review of "A Non-Conventional Platinum Drug against a Non-Small Cell Lung Cancer Line"

_molecules, 2023, doi:10.3390/molecules28041698_

Round 1
Reviewer 1 Report
I consider the subject of the manuscript entitled “A Non-Conventional Platinum Drug Against Lung Cancer” to be of moderate interest regarding publication in the Molecules journal.
As a general observation, the manuscript is well structured and well written, but it lacks in original studies and information related to metal-based drugs.
For once, it consists in a single studied complex, for which not even the synthesis and structural characterization is not provided as the complex was previously reported in another article. Although the microFTIR and microRaman spectroscopy studies are interesting, I do not believe they, by themselves, support the claims made by the authors regarding the mechanism of action of the complex. In my opinion, further in vitro studies are required, such as protein and lipid interaction assessments.
I am recommending publication after major revisions, as consistent studies need to be added for this article to be of real value.
Author Response
- Further in vitro studies are required.
The authors included the results of in vitro assays in lung non-tumorigenic cells (BEAS-2B cell line), complying with the reviewer's suggestion (and also with reviewer 2 comments, please see below). This allows to draw conclusions not only on the cytotoxic and antiproliferative effect of this complex against cancer cells but also on its impact on healthy cells (toxicity evaluation), the latter being of the utmost importance for a potential clinical application (maximizing anticancer activity while minimizing deleterious side effects).
Reviewer 2 Report
The authors J. D. Silva and co-workers submitted the manuscript entitled “A Non-Conventional Platinum Drug Against Lung Cancer” to the journal “Molecules” in order to be considered for publication as an “Article”.
The research work takes on the dinuclear platinum complex with putrescine linker. This complex has been described more than 30 years before, but is investigated in more detail in the present study. This covered evaluating the dose-dependent cytotoxicity using MTT/SRB assay after 24/48/72 h. Special focus was directed on analyses using microFTIR and micorRaman in order to prove interaction of the positively-charged and non-hydrolyzable (where has this been proven?) complex with the backbone of the DNA (clamp).
Thus, the topic of the manuscript generally meets the scope of the special issue “Metal-Based Drugs II”.
For the introduction, I would find it interesting/relevant to find what role putrescine has in the context of cancer. Is an anticancer activity described? What influence does it have on metabolism? etc.?
The cytotoxicity of the putrescine-containing dinuclear platinum complex is comparable to Cisplatin. However, what is an advantage? Is the new complex more active against Cisplatin-resistant cell lines (“may help to circumvent drug resistance mechanisms”)? Therefore, adducing a Cisplatin-resistant A549 cell line is highly recommended. How about the general toxicity? Maybe the new complex is less toxic against non-cancer cell lines? Generally, using a non-tumor cell line in the biological evaluation is state of the art. The authors are kindly asked to consider this in their study.
Basically, I find it difficult to emphasize the activity of the complex over lung cancer cells like that. On the one hand, only one single such cell line was used (abstract: “against human lung cancer cell lines”) - two or three other lung cancer cell lines would definitively be indicated. And what about other tumor types? They were not focused on at all. Thus, one cannot necessarily speak of significant activity against lung cancer (as the title implies).
“Comparing the average spectra of the untreated and [Pt2Put2(NH3)4]Cl4-treated samples, it was clear that the A549 cells were sensitive to the presence of the Pt(II) complex.” Is there really a difference in the spectra in Figure 3? The authors must please justify their conclusion on the sensitivity of the cells. Probably they should refer to their more detailed analysis below.
Personally, I do not find the research to be proving, just indicative. What also gives me pause for thought is that FBS is also used in cell culture with A542 cells. The amount of 10% are state of the art, but, however, not little. The FBS also contains a variety of proteins, amino acids, carbohydrates, hormones, vitamins, among others. It is well-known that platinum complexes show a high binding to FBS or other ingredients. Can this really be excluded/considered in the experimental setup? And then the complex is further dissolved in phosphate-containing saline. This does not make it easier in my opinion. Are the changes in the vibrations then exclusively due to the interaction with the DNA? I am unfortunately not sure…
The term “non-conventional bonds” seems a little bit strange to me. Maybe the authors can revise.
The authors are kindly asked to consider the positive charge of the dinuclear complex in case they do not express the four chloride counter-ions, i.e. the platinum complex is positively charged 4+.
Please introduce abbreviations at their first occurrence and then use them throughout the manuscript.
Please revise the manuscript with respect to typos.
Overall, I think the biological testing is far too poor. There is still significant potential for improvement to match the standard of the other publications at such an impact level. In addition, I find the evidence of the interaction with the DNA backbone only indicative, but not conclusive. The experimental setup does not necessarily make it easier to support the statements in my eyes. I am sorry for the authors, because I know it from my own experience, but I would not support a further processing of the manuscript in the current form. All the best!
Author Response
1) Special focus was directed on analyses using microFTIR and microRaman in order to prove interaction of the positively-charged and non-hydrolyzable (where has this been proven?) complex with the backbone of the DNA (clamp).
Although the complex [Pt2Put2(NH3)4]Cl4 has been synthesized by Farrell et al. more than 30 years ago [doi:10.1021/ic00317a005, ref 11 of the manuscript] at this time it was only intended as an intermediate for another polynuclear complex ([{trans-PtCl2(NH3)}2H2N(CH2)4NH2]). Hence, its potential antineoplastic properties were never reported up to this date.
The present manuscript reports the first such study focusing on the complex´s activity towards a human lung cancer cell line (A549), and comparing its effect on a non-malignant lung cell line. In particular, the specific structural characteristics of the chelate – absence of hydrolysable ligands and high positive charge – suggest a distinct mechanism of action relative to cisplatin and cisplatin-like drugs (which undergo intracellular chloride hydrolysis). The mode of action of this type of non-hydrolysable complexes – via electrostatic interaction with the negatively charged phosphate groups of DNA (phosphate clamps) – has been put forward, proved and reported by Farrell and coworkers for other similar Pt(II)-polyamine chelates (e.g. Triplatin [doi: 10.1039/c5cs00201j, 10.1016/j.ica.2019.118974, 10.1039/c4dt03237c, 10.1002/anie.201803448, 10.3389/fchem.2019.00307, refs 18-22 of the manuscript]).
2) For the introduction, I would find it interesting/relevant to find what role putrescine has in the context of cancer. Is an anticancer activity described? What influence does it have on metabolism?
The biological relevance of the biogenic polyamine putrescine, and its relationship to uncontrolled cell growth and neoplastic processes has been added to the manuscript/Introduction, as suggested.
3) Is the new complex more active against cisplatin-resistant cell lines? How about the general toxicity? Maybe the new complex is less toxic against non-cancer cell lines? Generally, using a non-tumor cell line in the biological evaluation is state of the art. The authors are kindly asked to consider this in their study.
Cytotoxicity assays in non-cancer lung cells (BEAS-2B cell line) have been included and discussed in the manuscript, and compared with reported data for cisplatin on the same cells. A lower toxicity has been observed for the dinuclear ([Pt2Put2(NH3)4]4+ agent on healthy versus cancer cells. Moreover, when comparing the current results with reported data for cisplatin [additional references added to the manuscript] [Pt2Put2(NH3)4]4+ shows to be less toxic towards lung non-tumour cells. These are promising results, aiming at an enhanced activity against cancer cells coupled to a lower effect (toxicity) on healthy cells, and is an added-value to a potential clinical application of [Pt2Put2(NH3)4]4+ (less deleterious side effects are to be expected relative to the conventional metallodrugs in current use).
Regarding the reviewer's suggestion of testing this Pt(II)-putrescine chelate in a cisplatin-resistant A549 cell line, this is foreseen for a near future as it is a highly relevant information attending to the severe clinical limitations associated to acquired resistance to Pt-drugs. In order to achieve this goal, however, this resistant cell line must be developed in the lab which is a time-consuming task (up to 6 months).
4) Is there really a difference in the spectra in Figure 3?
The difference spectra (treated cell–untreated cells (control), both FTIR and Raman) were included in the Figure (which is now Figure 4).
Nevertheless, the authors would like to provide the following explanation to the reviewer. Since FTIR and Raman spectra of these highly heterogeneous biological samples can be complex and difficult to interpret due to their similarity (as in this case as they are from the same cells), it is important to perform multivariate analysis techniques such as principal component analysis (PCA) in order to identify patterns in the data and to accurately recognise the spectral information that differs between the data groups. A simple analysis of the mean spectra might not completely reflect all the differences and variability found between the groups, and these differences are not usually apparent from a single variable analysis. Therefore, we have performed PCA, a powerful unsupervised technique that analyses all the acquired spectra and provides information about the major biomarkers that enable a class separation (if present). From the PCA scores (Fig. 5A, 5B and 5C) it is evident that a clear separation between the treated vs control cells is provided by both Raman and FTIR data. A detailed analysis of the loadings (Fig. 5D, 5E, and 5F) corresponding to the principal components that provide that separation, informs about the biochemical markers that differ between the two groups (control and drug-treated cells), as further specified along the manuscript. For clarity sake, we have added this explanation to lines 166-172 in the manuscript.
5) The FBS also contains a variety of proteins, amino acids, carbohydrates, hormones, vitamins, among others. It is well-known that platinum complexes show a high binding to FBS or other ingredients. Can this really be excluded/considered in the experimental setup? And then the complex is further dissolved in phosphate-containing saline. This does not make it easier in my opinion. Are the changes in the vibrations then exclusively due to the interaction with the DNA?
The complex was solubilized in PBS since it is poorly soluble in water. We considered more appropriate to solubilize it in a mixture of PBS:water (10% in saline, with which no interactions are expected) than in the commonly used water:DMSO solvent, since DMSO is recognizedly toxic to the cells (causing cell death). Moreover, reported similar studies (on drug effect on a biological matrix) use either saline or water:DMSO solutions.
Also, it should be emphazised that the interpretation of the results is based on a comparison between the data obtained for drug-free and drug-treated cells, all of these having been cultured and prepared under the same conditions, except for the absence or presence of the drug. Therefore, any possible effects of either the components of the culture medium or the solvent (saline solution), are ruled out when performing this comparative analysis. Hence, the spectral changes observed upon drug administration are justifiably due only to the interaction of the Pt-complex with the cellular components (e.g. DNA) and to the cellular response to the presence of this compound.
6) The term “non-conventional bonds” seems a little bit strange to me.
As explained above (item 1), the particular structural characteristics of the [Pt2Put2(NH3)4]Cl4 chelate – absence of hydrolyzable ligands and a 4+ charge – allow a distinct mechanism of action relative to cisplatin and cisplatin-like drugs (which undergo intracellular chloride hydrolysis): through electrostatic interaction with the negatively charged phosphate groups of DNA. This interplay with DNA has been reported by Farrell and coworkers as the phosphate clamp model, namely for TriplatinNC (charge=8+) [doi: 10.1039/c5cs00201j, 10.1039/c4dt03237c, refs 19 and 20 of the manuscript]. Also, the deviation from a conventional interaction with the target (direct metal coordination to one site at DNA) was also first described by these researchers [e.g. doi: 10.1039/c5cs00201j, ref 19 of the manuscript], referring to the fact that polynuclear complexes (with more than one metal center) are able to interact at more than one site at the DNA helix (in the case of phosphate clamp-type compounds with the phosphate moieties), thus allowing the formation of drug-DNA adducts which may be either intra- or inter-strand and long-range (since the distance between the metal centres in the complex is quite large, as defined by the length of the polyamine carbon chain, 4 CH2 groups in the case of putrescine). This differs from the conventional interplay between cisplatin (and cisplatin-based mononuclear drugs such as carboplatin and oxaliplatin) and DNA, which occurs by covalent binding of one metal center from the drugs and one base (usually a purine) in the nucleic acid, leading to adducts that cannot be long-range and are predominantly intra-strand (due to steric constraints).
7) The authors are kindly asked to consider the positive charge of the dinuclear complex in case they do not express the four chloride counter-ions, i.e. the platinum complex is positively charged 4+.
This has been considered and the text dully changed. The charge of the complex has also been included in Figure 1.
8) Please introduce abbreviations at their first occurrence and then use them throughout the manuscript.
All abbreviations have been checked and were explained in full at their first occurrence in the text. Additionally, the text has been revised for typos.
Round 2
Reviewer 1 Report
I appreciate the important additions and corrections made to the manuscript. I now consider it appropriate for publication.
Author Response
N.A.
Reviewer 2 Report
I thank the authors for pointing out and classifying their work with the anticancer studies of the complex, although its synthesis was described 30 years ago by Farrell. Indeed, I am aware of the matter. The authors state in their comments that Farrell has shown the "non-hydrolysable" property for other, albeit similar, complexes. However, direct evidence for the specific complex is lacking in this manuscript (the authors just state: "being expected"). It seems to me that this is missing in the literature (or did I just not find the corresponding reference) and therefore I think the proof is still to be provided.
The authors have included information about putrescine in the manuscript. That fits so far. Also, the authors provide in the manuscript what they mean by "non-coventional bonds". Formal inconsistencies (introduction of abbreviation, presentation of complex) have been addressed.
Regarding the highly recommended experiments using a resistant cell line of A549, the authors refer to the long time needed to generate such a cell line. Experiments of this kind could significantly increase the significance.
The general toxicity towards a non-tumorigenic cell line was performed using BEAS-2B as an example. However, the data presented in Table 1 raise the question whether this is the mean value of at least three independent investigations (SD/SE is missing then). This seems to me rather not to be the case. An independent repetition of the experiments is strongly recommended.
The authors do not even mention the biggest point of criticism of my previous comments in their response. I find this very unfortunate. Instead they simply changed the title without comment (which in my opinion still implies specific focus on lung cancer). I stand by my point that it is very misleading because a supposed special effect against lung cancer is suggested. Also the statement "against human lung cancer cell lines - both malignant and non-malignant" is misleading: what is meant by a non-malignant cancer cell line? In the conclusion the authors say "promising regarding the search for new metallodrugs and pharmacological targets against lung cancer". Who says that the complex cannot be active against other cancers?
The authors included the difference spectra (treated cell-untreated cells (control), both FTIR and Raman) to their manuscript. This makes their argumentation a bit more comprehensible. I still do not find it really proving. But maybe that is just my view.
Author Response
1) The authors state in their comments that Farrell has shown the "non-hydrolysable" property for other, albeit similar, complexes. However, direct evidence for the specific complex is lacking in this manuscript (the authors just state: "being expected"). It seems to me that this is missing in the literature (or did I just not find the corresponding reference) and therefore I think the proof is still to be provided.
The Pt(II) putrescine chelate [Pt2Put2(NH3)4]Cl4 does not contain any ligands that can undergo hydrolysis intracellularly. This does not really need proof (maybe our previous text when answering the reviewer was misleading). Indeed, hydrolysable (or leaving) ligands in this type of metal-based agents are usually chlorides or oxygen containing groups (such as cyclobutane dicarboxylic acid, in carboplatin for instance), which form labile bonds with the soft Pt(II) ion. Since Pt(II) is a soft cation, it yields labile bonds with hard ligands such as oxygen, and much stronger (non-labile) bonds with intermediate-to-soft ligands such as nitrogen. Also, chlorides are readily hydrolysed intracellularly due to the very low Cl- concentration in the cytoplasm (much lower than in the extracellular millieu). Therefore, [Pt2Put2(NH3)4]Cl4 displays stable bonds with the nitrogen atoms from both putrescine ligands (each metal centre being bound to 2 N´s from Put and 2 N´s from NH3 moieties) – none of these are readily hydrolysable inside the cell.
2) The general toxicity towards a non-tumorigenic cell line was performed using BEAS-2B as an example. However, the data presented in Table 1 raise the question whether this is the mean value of at least three independent investigations (SD/SE is missing then). This seems to me rather not to be the case. An independent repetition of the experiments is strongly recommended.
The [Pt2Put2(NH3)4]Cl4 was already under study against the non-tumourigenic cell line (BEAS-2B). Four independent measurements were performed for these studies (three being the minimum required for a reliable interpretation of the data) – this information was already included in the legend of Fig. 3 (n=4). SEM values were also added to Table 1.
3) The authors do not even mention the biggest point of criticism of my previous comments in their response. I find this very unfortunate. Instead they simply changed the title without comment (which in my opinion still implies specific focus on lung cancer). I stand by my point that it is very misleading because a supposed special effect against lung cancer is suggested. Also the statement "against human lung cancer cell lines - both malignant and non-malignant" is misleading: what is meant by a non-malignant cancer cell line? In the conclusion the authors say "promising regarding the search for new metallodrugs and pharmacological targets against lung cancer". Who says that the complex cannot be active against other cancers?
Firstly, the authors would like to emphasize that it was not their intention to overlook this concern of the reviewer. It was not the author´s aim to mislead the reader into a special effect of [Pt2Put2(NH3)4]Cl4 on lung cancer in general – only concerning a specific type of lung cancer (non-small cell A549), comparing with a particular non-tumourigenic cell line (BEAS-2B).
The title, which had not been changed before, was now altered with a view to convey a more accurate idea of the work reported in this manuscript.
In addition, all general mentions to “lung cancer” were clarified and specified along the text with regard to the A549 non-small cell line (as well as the BEAS-2B non-tumourigenic line).
The statement "against human lung cancer cell lines - both malignant and non-malignant" was substituted by “against human lung cell lines – both malignant (A549) and non-malignant (BEAS-2B)”.
The conclusion was also changed regarding the promising effect of the complex, focusing on this particular lung cancer type. What the authors meant was that the promising results currently obtained for the A549 cell line might allow us to be hopeful regarding a possible anticancer effect of this complex on other types of lung cancer. Naturally, this compound may also be active against other cancers apart from lung. However, the focus of this particular study was the A549 human lung cancer cell line. Further studies will follow (some are already underway on breast cancer) with a view to search for other antineoplastic activities of this Pt(II) binuclear chelate.